# Ecdysone-controlled nuclear receptor ERR regulates metabolic homeostasis in the disease vector mosquito *Aedes aegypti*

Dan-Qian Geng[1,2☯], Xue-Li Wang[1,2☯], Xiang-Yang Lyu[1,2], Alexander S. Raikhel[3,4]*, Zhen Zou[1,2]*

1 State Key Laboratory of Integrated Management of Pest Insects and Rodents, Institute of Zoology, Chinese Academy of Sciences, Beijing, China, 2 CAS Center for Excellence in Biotic Interactions, University of Chinese Academy of Sciences, Beijing, China, 3 Department of Entomology, University of California, Riverside, California, United States of America, 4 Institute for Integrative Genome Biology, University of California, Riverside, California, United States of America

☯ These authors contributed equally to this work.
* alexander.raikhel@ucr.edu (ASR); zouzhen@ioz.ac.cn (ZZ)

**Data Availability Statement:** The raw sequence data reported in this manuscript have been deposited with the National Genomics Data Center

## Abstract

Hematophagous mosquitoes require vertebrate blood for their reproductive cycles, making them effective vectors for transmitting dangerous human diseases. Thus, high-intensity metabolism is needed to support reproductive events of female mosquitoes. However, the regulatory mechanism linking metabolism and reproduction in mosquitoes remains largely unclear. In this study, we found that the expression of *estrogen-related receptor (ERR)*, a nuclear receptor, is activated by the direct binding of 20-hydroxyecdysone (20E) and ecdysone receptor (EcR) to the ecdysone response element (EcRE) in the *ERR* promoter region during the gonadotropic cycle of *Aedes aegypti* (named AaERR). RNA interference (RNAi) of *AaERR* in female mosquitoes led to delayed development of ovaries. mRNA abundance of genes encoding key enzymes involved in carbohydrate metabolism (CM)—*glucose-6-phosphate isomerase* (*GPI*) and *pyruvate kinase* (*PYK*)—was significantly decreased in *AaERR* knockdown mosquitoes, while the levels of metabolites, such as glycogen, glucose, and trehalose, were elevated. The expression of *fatty acid synthase* (*FAS*) was notably downregulated, and lipid accumulation was reduced in response to *AaERR* depletion. Dual luciferase reporter assays and electrophoretic mobility shift assays (EMSA) determined that AaERR directly activated the expression of metabolic genes, such as *GPI*, *PYK*, and *FAS*, by binding to the corresponding AaERR-responsive motif in the promoter region of these genes. Our results have revealed an important role of AaERR in the regulation of metabolism during mosquito reproduction and offer a novel target for mosquito control.

## Author summary

Following a blood meal in female mosquitoes, carbohydrate and lipid metabolism become activated, aligned with the high demands of reproductive requirements. Concurrently, the *estrogen-related receptor* (*AaERR*) mRNA level in *Aedes aegypti* rises dramatically. We

(NGDC, GSA: CRA012618), and are publicly accessible at https://ngdc.cncb.ac.cn/gsa.

**Funding:** This work was supported by National Natural Science Foundation of China Grants 32370522 (ZZ), 31802013 (XW), and 32370518 (XW), Open Research Fund Program of State Key Laboratory of Integrated Management of Pest Insects and Rodents (Chinese IPM2202) (ZZ), and NIH grant RO1 AI036959 (ASR). The funders had no role in the study design, in data collection, analysis or interpretation, in the decision to publish, or the preparation of the manuscript.

**Competing interests:** The authors have declared that no competing interests exist.

found that AaERR is regulated by 20-hydroxyecdysone (20E), which is the main regulatory hormone during the post blood meal (PBM) reproductive phase. The 20E receptor, ecdysone receptor (EcR), directly binds to the *AaERR* gene promoter region, activating transcription of this gene. 20E promotes glycolysis and glycogen catabolism, accelerating carbohydrate utilization. AaERR stimulates fatty acid synthesis and leads to lipid accumulation. We confirmed the direct regulatory relationship between AaERR and metabolism-related enzymes. Further investigation revealed that *AaERR* depletion inhibited the expression of *vitellogenin* (*Vg*) in female mosquitoes and disrupted the development of the ovaries, affirming the critical role of AaERR in 20E-dependent metabolism. These findings highlight the essential part of AaERR in maintaining metabolic homeostasis and promoting normal reproductive development in female mosquitoes.

## Introduction

Their obligatory blood-feeding behavior makes mosquitoes effective transmission vectors of human pathogens such as Plasmodium, dengue fever virus, Zika virus, yellow fever virus, and chikungunya virus. This leads to devastating consequences with nearly half a million people's deaths and millions of severe sicknesses annually [1–3]. However, due to the lack of effective vaccines and the high risk of the emergence of insecticide resistance, the control of mosquito-borne diseases is still facing enormous challenges [4]. Recently, the reduction of mosquito populations by controlling mosquito reproduction has become one of the promising strategies to prevent the spread of mosquito-borne diseases. Nutrients derived from vertebrate blood are essential for mosquitoes to fulfill their gonadotrophic cycles, and metabolism is required to be synchronized with these cycles. Extensive studies have reported that metabolism homeostasis plays a crucial role in insect reproduction [5–9]. Therefore, uncovering the regulatory mechanism of metabolic events during mosquito reproduction might lead to the development of novel methods for mosquito control.

Juvenile hormone (JH) and 20-hydroxyecdysone (20E), two main regulators governing insect development and reproduction, play key roles in metabolism [10–14]. In *Aedes aegypti*, these hormones affect metabolic homeostasis through their receptors, Methoprene-tolerant (Met)/Taiman (Tai) and Ecdysone receptor (EcR)/Ultraspiracle (USP), respectively [15–17]. Metabolites involved in carbohydrate metabolism (CM) and lipid metabolism (LM) are accumulated in the JH-controlled previtellogenic phase, but significantly decreased during the 20E-controlled vitellogenic phase [6,7]. Thus, a significant accumulation of metabolites during the JH-regulated PE phase was utilized later for vitellogenesis. In addition, JH and 20E signaling coordinately control the expression of *insulin-like peptides* (*ILPs*), which are involved in metabolic regulation during the reproductive cycle of *Ae. aegypti* [8]. In *Dipetalogaster maxima*, another hematophagous insect, JH signaling has been shown to play an important role in lipid storage in oocytes by regulating the expression of corresponding genes in both fat bodies (FBs) and ovaries [18]. A previous study showed that 20E affected the expression of genes controlling LM mainly during the post blood meal (PBM) phase in *Ae. aegypti* [7]. In addition, 20E inhibits glycolysis in the FB tissue during the molting of *Bombyx mori*, while JH exhibits the opposite effect [19]. Insulin and 20E signaling are crucial players in regulating the carbohydrate reserves of *B. mori* [14]. These observations indicate that the insect endocrine system coordinates development and reproduction with metabolism. However, the mechanism underlying endocrine control of metabolism still requires elucidation.

Nuclear receptors (NRs), a family of transcription factors, composed of the conserved zinc finger DNA binding domain (DBD) and C-terminal ligand-binding domain (LBD), have been shown to be important in regulating growth, development, and metabolism [9,20–22]. In mammals, estrogen-related receptors (ERRs) are members of the NR3B subfamily and are defined as orphan receptors due to the lack of their known ligands. They compose of three ERRs isoforms—ERRα, ERRβ, and ERRγ [23,24]—that are involved in cancer progression. ERRα and ERRγ are also linked to metabolic homeostasis [23,25–27]. Moreover, the target genes of ERR comprise genes related to energy metabolism [28–31]. Unlike mammals, however, only a single ERR protein is presented in *Drosophila*, and it is involved in regulating CM during the larval stage [32]. ERR is also implicated in controlling lipogenesis in young *Drosophila* adults [33]. However, the role of ERR has not yet been identified in important disease vectors such as mosquitoes.

In this study, we investigated the regulatory signaling pathway and biological function of ERR in the *Ae. aegypti* mosquito (named AaERR), an important vector of arboviral diseases such as dengue fever, yellow fever, Zika and chikungunya. In this work, we have shown that *AaERR* is expressed throughout the reproductive cycle, peaking at 36 h PBM; meanwhile, the titer of 20E remained at a high level during 18–24 h PBM, indicating that the expression of *AaERR* might be associated with the accumulation of 20E [6,34]. Moreover, we have demonstrated that upon binding with 20E, the EcR/USP complex directly regulates *AaERR* expression via binding to the ecdysone response element (EcRE) in the promoter region of the *AaERR* gene. Depletion of *AaERR* in mosquitoes led to severely delayed development of the ovary and disturbed the metabolism homeostasis. We further showed that AaERR was essential for the transcriptional expression of CM and LM genes and regulated these genes by directly binding to their regulatory regions. Thus, the findings presented here elucidate the key role of AaERR in 20E-dependent metabolism and oocyte maturation. They extend the understanding of 20E regulation in mosquito reproduction and provide a basis for the future development of the mosquito population control approaches based on the manipulation of mosquito energy metabolism.

## Results

### 20E regulates the expression of *AaERR* through its receptor EcR in *Ae. aegypti* female mosquitoes

As in *Drosophila*, only a single *AaERR* gene is present in the genome of *Ae. aegypti*. We first determined the temporal distribution of *AaERR* mRNA in the FB of female mosquitoes throughout the first gonadotrophic cycle. FB samples were collected from female mosquitoes at ten time points—6 and 72 hours (h) post eclosion (PE); and 6, 18, 24, 36, 48, 60, 72, and 96 h PBM—and the *AaERR* mRNA abundance was detected using quantitative reverse transcription PCR (qRT-PCR). The *AaERR* mRNA level was low between 6 h and 72 h PE, as well as at the beginning of the PBM phase. It rose sharply after 24 h PBM, reaching a peak at 36 h PBM, then precipitously declining by 48 h PBM. The *AaERR* mRNA level at 72–96 h PBM was like that at 72 h PE (Fig 1A). Recent studies have revealed that the hormones 20E play a pivotal role during vitellogenesis of mosquitoes [13]. The fluctuations of 20E post blood meal are depicted at the top of Fig 1A, effectively illustrating the correlation between 20E titers and *AaERR* [34]. The mRNA expression of *AaERR* in mosquitoes shows a distinct pattern: it rises slowly within the first 24 h PBM, then sharply increases, reaching its peak at 36 h PBM. This pattern suggests that *AaERR* transcription activation is closely dependent on the 20E titer, with significant activation occurring only when the 20E level is substantially high. Furthermore, our prior research indicated that mosquitoes exhibit a heightened metabolic activity at 36 h PBM [6]. This is

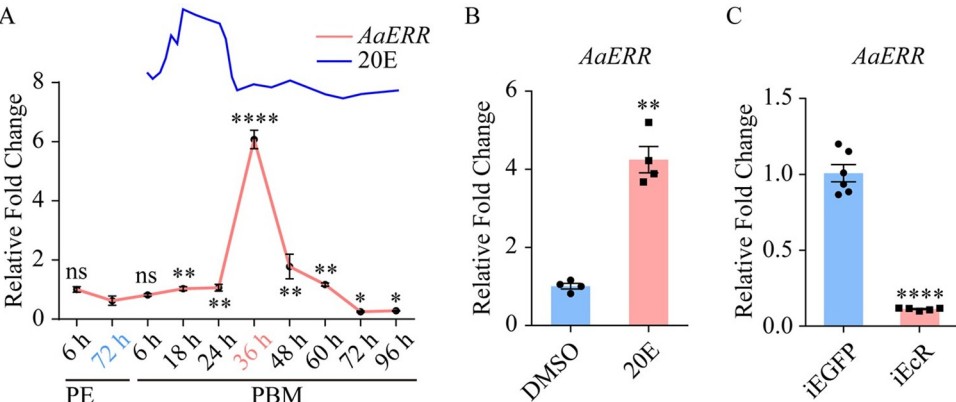

**Fig 1. Expression dynamics and hormone regulation of *AaERR* in the adult female *Ae. aegypti*.** (A) Relative *AaERR* mRNA levels in the FB were examined across various developmental stages. The titers of 20E post blood meal phase was schematically shown at the top, which was modified from the published data [34]. (72 h PE was used as a control. Statistical tests were shown in S3 Table; at least three biological replicates). Error bars are shown as mean ± SEM. (B) The *in-vitro* FB culture assay demonstrated the 20E activation effect on *AaERR* expression (Two-tailed Welch's t test: **p = 0.0018; at least three biological replicates). Error bars are shown as mean ± SEM. (C) The effect of *EcR* depletion on *AaERR* expression in the FB of female mosquitoes compared with that in the *enhanced green fluorescent protein* (*EGFP*) knockdown groups. Mosquitoes were given a blood meal three days after *EcR* knockdown, and FBs were collected to measure the *AaERR* transcript at 36 h PBM. (Two-tailed Welch's t test: ****p < 0.0001; at least three biological replicates). Error bars are shown as mean ± SEM.

further supported by the observation that several metabolism-related genes, including *HR38*, *E93*, and *AaE74A*, which are regulated by 20E, also reach their peak expression at 36 h PBM [5,16,35,36].

JH plays an important role in the PE phase of mosquitoes [13]. To understand a possible role of JH in regulating the *AaERR* gene expression, we conducted an RNA interference (RNAi) knockdown experiment for the JH receptor Met (iMet). Results showed that *AaERR* mRNA level increased in the FBs during the PE phase suggesting that JH-Met may actively inhibit *AaERR* expression (S1A and S1B Fig). Given that the mRNA levels of *AaERR* is minimal during the PE phase but significantly elevated during the PBM phase (Fig 1A) when the 20E titer rises sharply, we propose that AaERR predominantly functions in regulating events during the PBM phase under the control of 20E that is a pivotal hormone driving reproductive events and gene expression post-blood meal. Thus, we explored a possible involvement of this hormone in regulating *AaERR* gene expression. As the first step, we performed an *in-vitro* FB culture experiment, as previously described [37,38]. Isolated FBs from female mosquitoes at 72 h PE were incubated for 10 h in the culture medium containing 2 μM 20E. The control FBs were incubated for the same time in culture medium without 20E. We found that under these conditions, 20E significantly induced the *AaERR* expression (Fig 1B). Next, we conducted *EcR* double-stranded RNA (dsRNA) knockdown (S1C Fig). The *AaERR* transcript level decreased after knocking down *EcR* (Fig 1C). This result indicates that 20E is involved in the regulation of *AaERR* expression through its receptor EcR.

## The 20E-bound EcR/USP directly interacts with the *AaERR* gene regulatory region

After establishing that the *AaERR* gene expression is regulated by 20E, we investigated whether EcR directly regulates the *AaERR* gene. First, the upstream regulatory region of *AaERR* (nt -2101 to +236) was analyzed in JASPAR (https://jaspar.genereg.net/). We identified a potential EcR binding motif (GTGCTCAATGAACTT) within this *AaERR* regulatory region. To explore

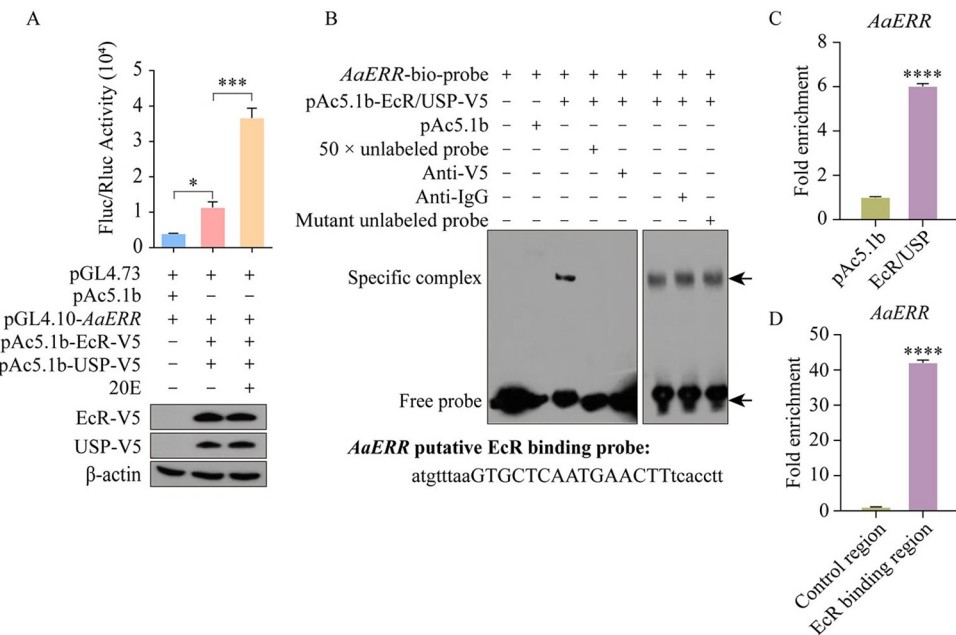

**Fig 2. Transcriptional regulation of *AaERR* by EcR/USP bound to 20E.** (A) Dual luciferase reporter assay was performed after co-transfecting pGL4.10-*AaERR* (nt -2101 to +236) into S2 cells with pAc5.1b empty vector or pAc5.1b-EcR-V5 and pAc5.1b-USP-V5 with or without 20E (2 μM). pGL4.73 vector was used as an expression control in this assay (Two-tailed Welch's t test: *p = 0.0176; Two-tailed Unpaired t test: ***p = 0.0002; at least three biological replicates). Error bars are shown as mean ± SEM. Western blot was used to demonstrate the expression of EcR-V5 and USP-V5 proteins in S2 cells. β-Actin was used as a loading control. (B) EMSA was used to show the specific binding between the *AaERR* probe and EcR/USP-V5 complex from Aag2 cell nuclear extract. Specific complex band and free probe are indicated by the arrow. Nuclear extract of Aag2 cells transfected with pAc5.1b empty vector was used as a control. The unlabeled *AaERR* probe and its corresponding mutant probe, both in 50 × molar excess, were employed to verify the binding site's specificity. The anti-V5 antibody was utilized to confirm the binding specificity of antibody with the EcR/USP-V5 complex in protein-probe binding mixtures, with the binding efficacy demonstrated using the anti-IgG antibody. Specific complex band and free probe are indicated by arrows. The sequence of the *AaERR* probe is displayed under the EMSA results. (C) ChIP-qPCR analysis was performed using anti-V5 and anti-IgG antibodies to assess the enrichment of *AaERR* promoter fragments in Aag2 cells with overexpressed EcR/USP-V5 proteins. The pAc5.1b vector was used as a control in cells and was similarly tested with both antibodies. The graph illustrated the relative fold enrichment of the *AaERR* promoter fragments in comparison to the control. (Two-tailed Unpaired t test: ****p < 0.0001; three biological replicates). Error bars are shown as mean ± SEM. (D) qPCR analysis of DNA precipitated post-ChIP with anti-V5 antibody, assessing the enrichment of the *AaERR* regulatory region against a control region within its coding sequence. (Two-tailed Unpaired t test: ****p < 0.0001; three biological replicates). Error bars are shown as mean ± SEM.

the effect of EcR on *AaERR* expression, we utilized the dual luciferase reporter assay. Co-expression of the plasmid containing the *AaERR* regulatory region with the putative EcRE binding site together with EcR-V5 and USP-V5 resulted in a 2.89-fold increase in relative luciferase activity in comparison to the empty vector. Moreover, when 20E was added, the luminescence intensity increased to 3.23-fold compared to co-expressing EcR-V5 and USP-V5 without 20E (Fig 2A). The presence of EcR-V5 and USP-V5 fusion proteins in S2 cells was confirmed by western blotting (Fig 2A). These results suggest that 20E significantly amplifies the effect of EcR/USP on *AaERR* transcription.

Electrophoretic mobility shift assay (EMSA) was next conducted to identify the binding of the EcR/USP complex to the *AaERR* gene promoter region. The nuclear extract from the mosquito Aag2 cell line transfected with pAc5.1b-EcR-V5 and pAc5.1b-USP-V5 was used for EMSA. The shift band was evident after the biotin-labeled *AaERR* probe was combined with the nuclear extract. This band was not visible after the addition of 50 × molar excess of the

unlabeled specific *AaERR* probe to the binding reaction, whereas the addition of 50 × molar excess of the unlabeled mutant probe did not have this effect. Moreover, the band shift disappeared from a gel when the reaction mixture was pre-incubated with the anti-V5 antibody for 1–2 h at 4°C. In contrast, when the anti-IgG antibody was added to the mixture instead of the anti-V5 antibody, the gel band remained indicating that this antibody did not disrupt the specific probe-protein interaction (Fig 2B). These EMSA experiments indicate that existence of specific binding of EcR with the regulatory region of the *AaERR* gene.

Subsequently, we conducted the chromatin immunoprecipitation (ChIP) reaction coupled with quantitative polymerase chain reaction (qPCR) to evaluate the binding between *AaERR* and EcR. We discovered that in Aag2 cells overexpressing EcR/USP-V5, the *AaERR* gene regulatory region DNA fragments obtained through ChIP, showed significant enrichment compared to the control groups from cell transfected with the pAc5.1b vector, confirming the presence of EcR binding motifs in the promoter region of *AaERR* (Fig 2C). qPCR analysis of DNA fragments after ChIP with an anti-V5 antibody showed that the enrichment of *AaERR* regulatory region was significantly higher than that in the control region within the coding sequence (Fig 2D). This further confirms the direct binding of EcR to the promoter region of *AaERR*.

## The reproductive development of female *Ae. aegypti* is affected by AaERR

Previous studies have shown that 20E is a key hormone that regulates the development of female mosquitoes during the PBM phase [39]. ERR has also been reported to affect the reproductive development in both mammals and insects [29,40,41]. To explore the effect of AaERR in female mosquito reproduction, the 24 h PE mosquitoes were injected with dsRNA, with either *AaERR* dsRNA or *EGFP* dsRNA. *EGFP* as a control. They were fed with blood at 72 h PE and examined at 36 h PBM. The *AaERR* mRNA level after RNAi of *AaERR* (iAaERR) significantly decreased (S1D Fig). We found that the development of ovaries was significantly affected in iAaERR mosquitoes compared with *EGFP* RNAi-depleted (iEGFP) mosquitoes (Fig 3A). The ovarian follicles of iAaERR females were smaller (0.355 mm on average) than those of iEGFP individuals (0.394 mm on average) (Fig 3B). *AaERR* RNAi led to a significant reduction in egg numbers (109 on average) compared with *EGFP* RNAi (123 on average) (Fig 3C). Moreover, the level of *vitellogenin* (*Vg*) mRNA was also reduced compared with the

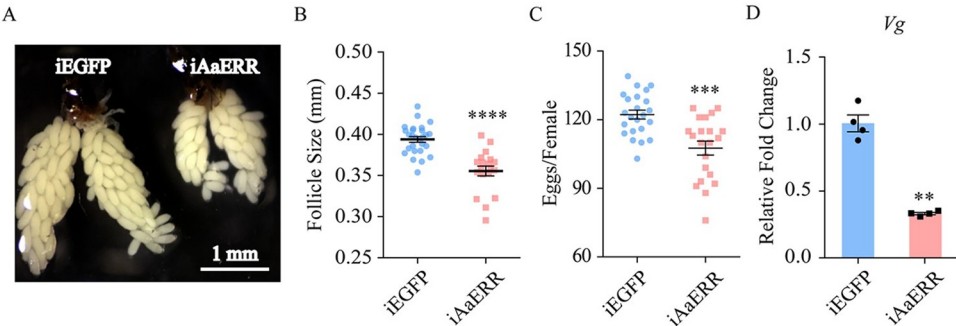

**Fig 3. AaERR affects reproductive development.** (A) The development of ovarian follicles was suppressed in *AaERR*-knockdown *Ae. aegypti* compared with that in iEGFP mosquitoes (error bar: 1 mm). The images were captured under a NIKON A1 microscope. (B-C) Statistical analysis was performed on the average follicle size (B) and egg deposition (C) of iAaERR and iEGFP mosquitoes (Two-tailed Mann-Whitney test: ****p < 0.0001; Two-tailed Mann-Whitney test: ***p = 0.0004; at least three biological replicates). Error bars are shown as mean ± SEM. (D) The relative mRNA levels of *Vg* at 36 h PBM were examined in mosquitoes with dsEGFP or dsAaERR (Two-tailed Welch's t test: **p = 0.0015; at least three biological replicates). Error bars are shown as mean ± SEM.

control group (Fig 3D). These results indicate that AaERR is important for maintaining the normal reproductive development of female mosquitoes.

## The CM and LM of *Ae. aegypti* females are affected by AaERR

In female mosquitoes, metabolism is tightly linked to reproduction. Recent studies have shown that ERR plays an important role in regulating metabolism [23]. However, the molecular mechanism of AaERR on metabolic genes remains to be elucidated. Therefore, we investigated the effect of AaERR on the metabolic process of female *Ae. aegypti*. The FB of iAaERR mosquitoes at 36 h PBM had higher contents of glucose, fructose, and trehalose, detected by gas chromatography-mass spectrometry (GC-MS) (Fig 4A). The results of periodic acid Schiff (PAS) staining showed that the FB of iAaERR mosquitoes displayed a significantly higher glycogen level (Fig 4B). Similarly, the results of colorimetric measurement techniques also demonstrated significantly higher glycogen content in the FB of iAaERR mosquitoes at 36 h PBM (Fig 4C). It indicates that AaERR could promote glycogen catabolism. The LM process was also found to be regulated by AaERR. Nile red staining results showed that the size (area in $\mu m^2$) and density of lipid droplets after *AaERR* knockdown were affected (Fig 4D and 4E). The levels of triacylglycerol (TAG), the main component in lipid droplets, also obviously decreased at 36 h PBM in the FB of iAaERR mosquitoes measured by colorimetry (Fig 4F). These results indicate that AaERR plays a key role in both CM and LM controlling carbohydrate metabolite consumption and lipid accumulation.

## Transcriptomic analysis of *AaERR* RNAi knockdown mosquitoes

To decipher the mechanism by which AaERR regulates metabolism, we conducted RNA-sequencing (RNA-Seq). To compare gene expression differences between iAaERR and iEGFP female mosquito samples at 36 h PBM, genes with p value <0.05 and fold change >1.5 or <0.667 were considered as differentially expressed genes (DEGs). As shown in the volcano plot, a total of 118 upregulated DEGs (orange) and 366 downregulated DEGs (blue) were detected (Fig 5A). After conducting the KEGG pathway enrichment analysis of DEGs in iAaERR mosquitoes, all the annotated CM and LM genes were identified. The transcriptomic changes in these genes and the predicted binding motifs in their promoters are shown in the S4 Table. This demonstrates that more DEGs were downregulated after *AaERR* depletion. KEGG pathway enrichment analysis was performed to characterize the functions of downregulated gene cohorts, indicating that the genes involved in CM and LM are mainly distributed in the downregulated rather than upregulated cohorts (Figs 5B and S2). The genes that matched glycolysis/gluconeogenesis, pentose phosphate pathway and fatty acid metabolism are shown in detail using Sankey dot pathway enrichment (Fig 5C). The KEGG pathway enrichment analysis of the downregulated gene repertoire in glycolysis revealed diminished expression levels of key enzymes in iAaERR mosquitoes, signifying inhibition of the glycolysis pathway. Similarly, the analysis of downregulated DEGs in fatty acid metabolism indicates a greater number of genes in fatty acid synthesis than in degradation. Consequently, glycolysis and fatty acid synthesis are inhibited in the FB of iAaERR mosquitoes, leading to an elevation in carbohydrate metabolites and a reduction in lipid content.

## AaERR directly regulates key enzymes in the CM pathway

We first measured the mRNA levels of CM key enzyme genes in the female FBs at 36 h PBM. Consistent with the metabolic phenotype observed above, the mRNA levels of four representative genes, including *glucose-6-phosphate isomerase* (*GPI*), *pyruvate kinase* (*PYK*), *phosphofructokinase* (*PFK*), and *phosphoglucomutase* (*PGM*), were significantly lower in iAaERR

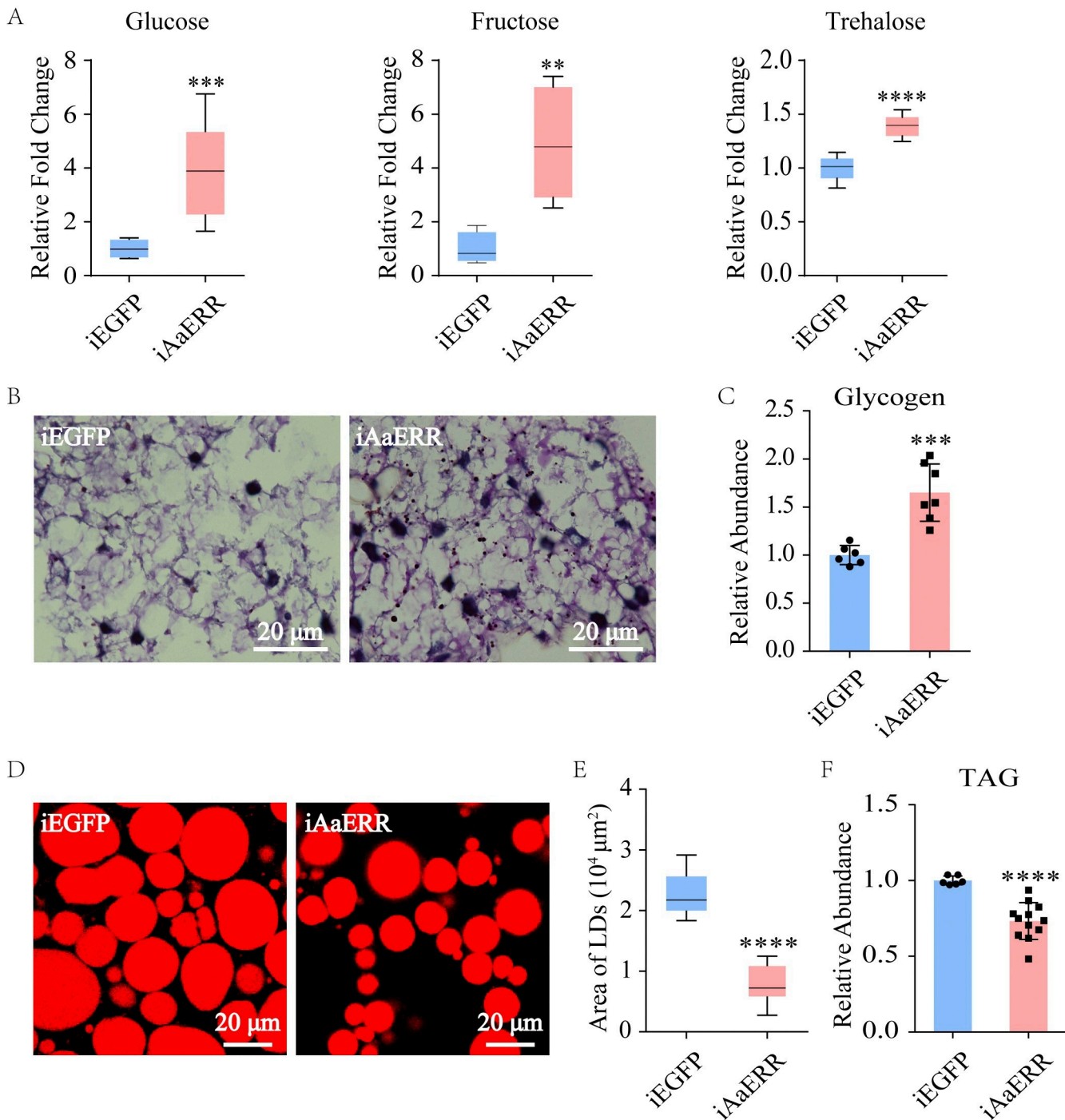

**Fig 4. Effect of *AaERR* depletion on the CM and LM pathways of *Ae. aegypti*.** (A) Glucose, fructose, and trehalose contents in the FB of iAaERR and iEGFP female mosquitoes were detected using GC-MS (Two-tailed Welch's t test: ***p = 0.0005; Two-tailed Unpaired t test: **p = 0.0054; Two-tailed Unpaired t test: ****p < 0.0001; at least three biological replicates). Error bars are shown as mean ± SEM. (B) The content of glycogen in the FB of female mosquitoes was visualized using periodic acid Schiff (PAS) staining (error bars: 20 μm). (C) The levels of glycogen were measured in the FB of iAaERR and iEGFP mosquitoes (Two-tailed Welch's t test: ***p = 0.0008; at least three biological replicates). Error bars are shown as mean ± SEM. (D) Lipid droplets in the FBs of iAaERR and iEGFP mosquitoes were detected by Nile red staining (error bars: 20 μm). (E) Quantification of the lipid droplets sizes of Fig 4D (Two-tailed Unpaired t test: ****p < 0.0001). Error bars are shown as mean ± SEM. (F) Levels of triacylglycerol were measured in iAaERR and iEGFP mosquitoes (Two-tailed Welch's t test: ****p < 0.0001; at least three biological replicates). Error bars are shown as mean ± SEM. All samples described were obtained at 36 h PBM. Measurements in (C) and (F) were normalized to the total proteins, and iAaERR treatments were further normalized to iEGFP treatments.

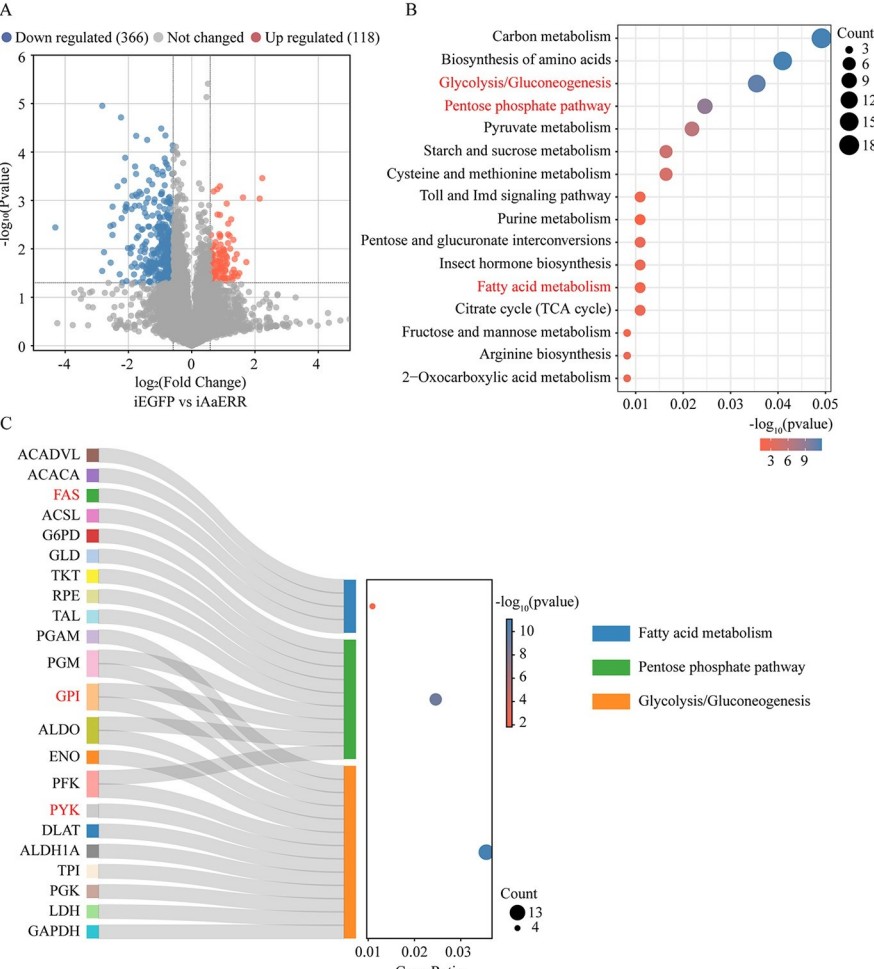

**Fig 5. Comparison of the transcriptome data between iEGFP and iAaERR mosquitoes.** (A) Volcano plot analysis was employed to illustrate the DEGs after *AaERR* knockdown. The downregulated (blue) and upregulated (orange) genes with a fold change of 1.5 and p <0.05 are shown. Gray plots represent genes with no significant differences. (B) KEGG pathway enrichment analysis was conducted to identify the DEGs that were downregulated. The metabolic pathways of interest are highlighted in red. (C) Gene Set Enrichment Analysis Sankey and dot plot were performed to analyze significantly downregulated DEGs. The abscissa represents the gene ratio, the ordinate represents different metabolic pathways, and the dot size displays the counts of enriched genes. Gene ratio is defined as the ratio of the number of genes enriched from each category to the total number of genes analyzed in the KEGG pathway enrichment analysis. Three categories are shown in (C). Genes used in further analysis are labeled red. ACADVL, very long chain acyl-CoA dehydrogenase; ACACA, acetyl-CoA carboxylase/biotin carboxylase 1; FAS, fatty acid synthase; ACSL, long-chain acyl-CoA synthetase; G6PD, glucose-6-phosphate 1-dehydrogenase; GLD, glucose 1-dehydrogenase; TKT, transketolase; RPE, ribulose-phosphate 3-epimerase; TAL, transaldolase; PGAM, phosphoglycerate mutase 2; PGM, phosphoglucomutase; GPI, glucose-6-phosphate isomerase; ALDO, fructose-bisphosphate aldolase; ENO, enolase; PFK, 6-phosphofructokinase; PYK, pyruvate kinase; DLAT, dihydrolipoamide acetyltransferase; ALDH1A, retinal dehydrogenase; TPI, triosephosphate isomerase; PGK, phosphoglycerate kinase; LDH, L-lactate dehydrogenase; GAPDH, glyceraldehyde 3-phosphate dehydrogenase.

mosquitoes (Figs 6A and S3A). There is no doubt that AaERR can regulate the mRNA levels of key enzyme genes in the CM pathway, but the more detailed mechanism still needs to be further explored. Then, we compared the target genes binding sequences of ERR among different insect species and found that AaERR binding motif was conserved (Fig 6B). This conserved sequence was also found in the promoter regions of all the CM key enzyme genes explored in this article (S2 Table).

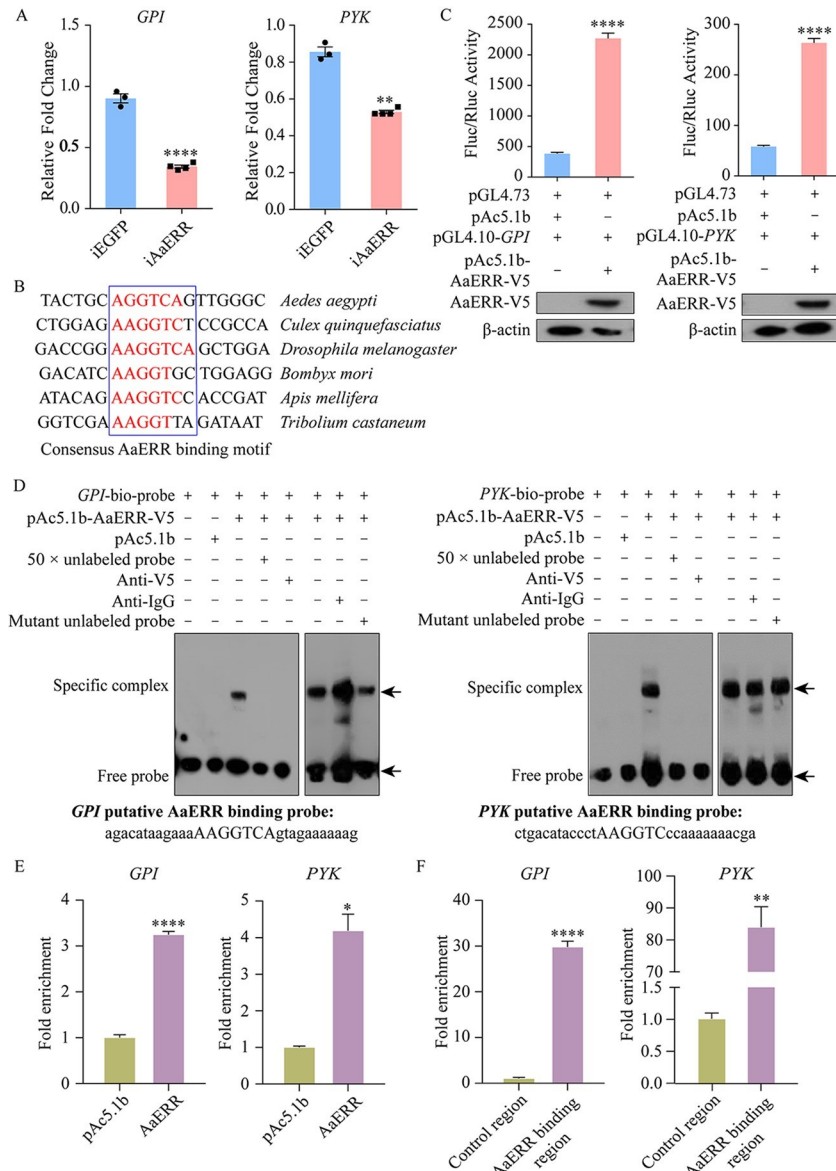

**Fig 6. AaERR directly regulates the expression of key enzymes in the CM pathway.** (A) The relative mRNA levels of *GPI* and *PYK* at 36 h PBM were affected by the depletion of *AaERR* (Two-tailed Unpaired t test: ****p < 0.0001; Two-tailed Welch's t test: **p = 0.0063; at least three biological replicates). Error bars are shown as mean ± SEM. (B) Multiple sequence alignments were performed, utilizing the promoter region of the target genes from various insect species. A conserved AaERR binding motif is shown in the blue box. (C) The role of AaERR in regulating *GPI* and *PYK* expression was verified using dual luciferase reporter assay. In this experiment, pGL4.10-*GPI* (nt -808 to +313) or pGL4.10-*PYK* (nt -843 to -135) were co-transfected with pAc5.1b-AaERR-V5. pAc5.1b empty vector was used in the control group (Two-tailed Unpaired t test: ****p < 0.0001; at least three biological replicates). Error bars are shown as mean ± SEM. Western blotting showed the overexpression of AaERR-V5 protein in S2 cells. β-actin was used as a loading control. (D) The EMSA experiments were used to confirm the specific binding between nuclear extract of Aag2 cells transfected with AaERR-V5 protein and *GPI* and *PYK* probes. Nuclear extract of Aag2 cells transfected with pAc5.1b empty vector was used as a control. The anti-V5 antibody was employed to confirm the binding of AaERR-V5 protein to the biotin-labeled probe, while the anti-IgG antibody served to demonstrate the specific interaction between the anti-V5 antibody and the AaERR-V5 protein. To ascertain the binding motifs' specificity, we used the unlabeled *GPI* and *PYK* probes, and their mutant forms, each in 50 × molar excess. Specific complex band and free probe are indicated by arrows. The sequences of *GPI* and *PYK* probes were displayed under each EMSA result. (E) ChIP-qPCR analysis was carried out using anti-V5 and anti-IgG antibodies to evaluate the enrichment of promoter fragments for *GPI* and *PYK* in Aag2 cells transfected with the pAc5.1b-AaERR-V5 construct. Cells transfected with the pAc5.1b vector alone served as a control and underwent the same antibody testing. The resulting graph displays the fold

enrichment of the *GPI* and *PYK* promoter fragments (Two-tailed Unpaired t test: ****p < 0.0001; Two-tailed Welch's t test: *p = 0.0192; three biological replicates). Error bars are shown as mean ± SEM. (F) qRT-PCR was used to analyze the precipitated DNA to compare the enrichment of the regulatory region and a control region in the coding sequences of *GPI* and *PYK* after ChIP with the anti-V5 antibody. (Two-tailed Unpaired t test: ****p < 0.0001; Two-tailed Welch's t test: **p = 0.0058; three biological replicates). Error bars are shown as mean ± SEM.

To identify whether AaERR directly affects the mRNA levels of CM genes, we used the dual luciferase reporter assay. The upstream regulatory regions of *GPI* (nt -808 to +313) and *PYK* (nt -843 to -135) were analyzed in the JASPAR database (https://jaspar.genereg.net/). We found that a putative ERR binding motif (AAGGTCA) was present in their promoter regions. We subcloned the cDNA sequence of *AaERR* into a pAc5.1b/V5 vector to express AaERR-V5 fusion protein in S2 cells. The promoter regions of *GPI* and *PYK* were amplified and cloned to the separate pGL4.10 vector. We then co-transfected the plasmid pGL4.10-*GPI*-promoter or pGL4.10-*PYK*-promoter with pAc5.1b-AaERR-V5 into *Drosophila* S2 cells, along with the empty vector pAc5.1b as the control. The luminescence intensity in experimental groups significantly increased 5.93-fold (*GPI*) and 4.54-fold (*PYK*). The expression of AaERR-V5 protein in S2 cells was confirmed by western blotting (Fig 6C).

The EMSA experiment was then used to verify the AaERR binding specificity to the identified promoter regions of *GPI* and *PYK* that contained putative ERR binding sites. The experiment was conducted in a similar manner as for testing interaction of EcR and *AaERR* (see above). The band was apparent after the biotin-labeled *GPI* or *PYK* probes was combined with the nuclear extract containing plasmid. It disappeared after adding 50 × molar excess of unlabeled specific probes. However, the band remained after the 50 × molar excess unlabeled mutant probe was added to the binding reaction. To further confirm the presence of AaERR-V5 in the nucleoprotein-probe complexes, the anti-V5 antibody and nuclear extract were incubated together on ice for 1–2 h before the binding experiment, then the band shift disappeared. However, the addition of anti-IgG antibody did not affect the shift band indicating that this antibody did not disrupt the specific probe-protein interaction (Fig 6D). These results show that AaERR binds to a specific region in both the *GPI* and *PYK* promoters. Thus, AaERR affects the CM by directly regulating the *GPI* and *PYK* mRNA levels.

Next, we conducted the ChIP assay coupled with qPCR to assess binding between regulatory regions of certain CM genes and AaERR. In Aag2 cells overexpressing AaERR-V5, DNA fragments in regulatory regions of *GPI*, *PYK*, *PFK*, and *PGM*, obtained through ChIP, were highly enriched compared to the control groups from cells transfected with the pAc5.1b vector (Figs 6E and S3C). At the same time, we performed qPCR analysis of coding regions of *GPI*, *PYK*, *PFK* and *PGM* after ChIP with anti-V5 antibody, and found no enrichment (Figs 6F and S3E). The results verify the binding of AaERR to the regulatory regions of *GPI*, *PYK*, *PFK* and *PGM*.

## AaERR directly regulates key enzymes in the LM pathway

Our transcriptome analysis revealed four DEGs related to fatty acid metabolism in *AaERR* RNAi knockdown mosquitoes (Fig 5C). Despite the downregulation of all four genes in these mosquitoes, three are involved in fatty acid synthesis, and one, *very long-chain acyl-CoA dehydrogenase* (*ACADVL*), is associated with fatty acid degradation. This pattern indicates a possible pronounced inhibition of the fatty acid synthesis pathway compared to the degradation pathway in iAaERR mosquitoes, leading to an overall reduction in lipid levels. Therefore, the three enzymes associated with fatty acid synthesis attracted a particular interest in our study.

We measured the transcriptional abundance of some key enzymes of the LM pathway in female mosquito FBs using qRT-PCR. The mRNA levels of *fatty acid synthase* (*FAS*), *ACSL*

(*long-chain acyl-CoA synthetase*), and *ACACA* (*acetyl-CoA carboxylase/biotin carboxylase 1*) were significantly lower in iAaERR mosquitoes at 36 h PBM (Figs 7A and S3B). Next, we analyzed the upstream regulatory region of *FAS* (nt -1913 to -6) using JASPAR (https://jaspar.genereg.net/) and identified a putative AaERR binding motif (AAGGTCA). We then amplified and subcloned the *FAS* promoter region into the pGL4.10 vector. pAc5.1b-AaERR-V5 and pGL4.10-*FAS*-promoter were co-transfected into *Drosophila* S2 cells. pGL4.10-*FAS*-promoter co-transfected along with pAc5.1b empty vector were served as the control. The intracellular luminescence intensity was 3.76-fold higher after co-transfection with pAc5.1b-AaERR-V5 and pGL4.10-*FAS*-promoter than the control. The expression of AaERR-V5 protein in S2 cells was confirmed using western blot (Fig 7B).

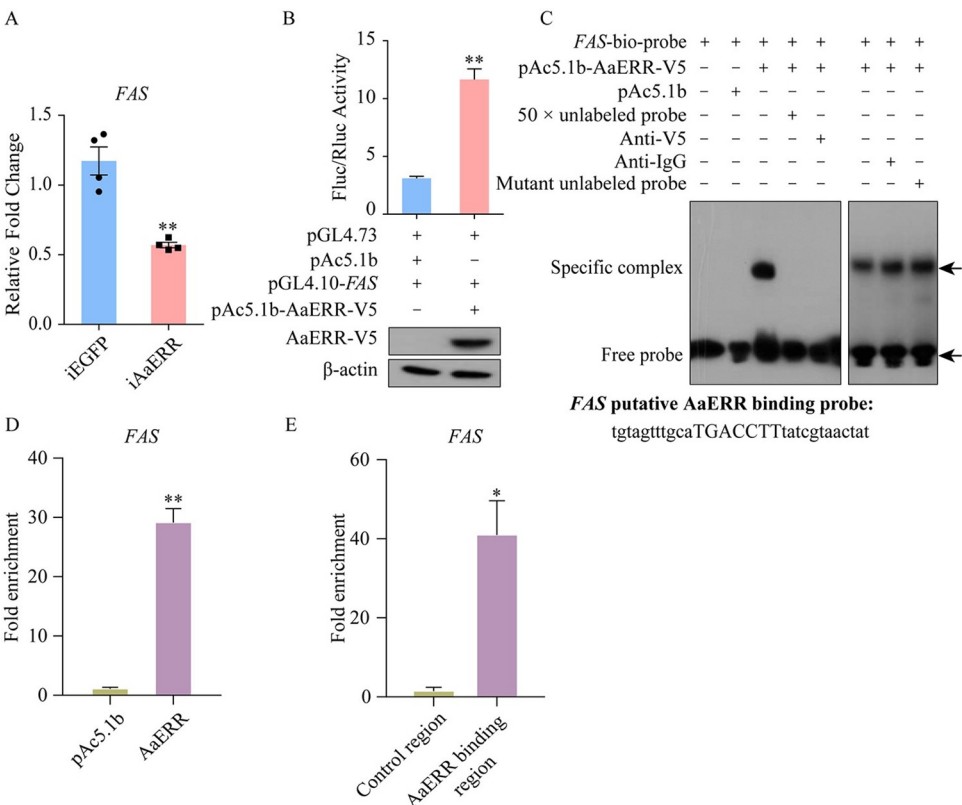

**Fig 7. AaERR directly regulates the expression of key enzymes in the LM pathway.** (A) The effect of *AaERR* depletion on the expression of *FAS* at 36 h PBM was evaluated and compared with that in iEGFP mosquitoes (Two-tailed Welch's t test: **p = 0.0080; at least three biological replicates). Error bars are shown as mean ± SEM. (B) Dual luciferase reporter assay was used to verify the regulation of *FAS* by AaERR. pGL4.10-*FAS* (nt -1913 to -6) co-transfected with pAc5.1b empty vector or pAc5.1b-AaERR-V5 in S2 cells, and western blotting showed the overexpression of AaERR-V5 protein (Two-tailed Welch's t test: **p = 0.0021; at least three biological replicates). Error bars are shown as mean ± SEM. (C) EMSA experiment confirmed the specific binding site of AaERR within the promoter region of *FAS* gene. pAc5.1b-AaERR-V5 was transfected into Aag2 cells, then the nuclear protein was extracted to incubate with the *FAS* biotin-labeled probe. pAc5.1b empty vector was transfected into Aag2 cells as the control group. The anti-V5 antibody was used to verify the binding of AaERR-V5 to the biotin-labeled probe, with the anti-IgG antibody further confirming the specific binding between the anti-V5 antibody and the protein. The specificity of the binding site was demonstrated using the unlabeled *FAS* probe and its mutant form, both in 50 × molar excess. Specific complex band and free probe are indicated by arrows. The sequence of *FAS* probe was displayed under the EMSA results. (D) ChIP-qPCR analysis with AaERR-V5 overexpressed Aag2 cells using anti-V5 and anti-IgG antibodies. As a control, cells transfected with pAc5.1b vector were also tested with the same two antibodies. The enrichment of *FAS* promoter fragments was analyzed by ChIP-qPCR. (Two-tailed Welch's t test: **p = 0.0065; three biological replicates). Error bars are shown as mean ± SEM. (E) ChIP using an anti-V5 antibody followed by qPCR quantified the enrichment of the *FAS* regulatory region against a control region in its coding sequence. (Two-tailed Welch's t test: *p = 0.0434; three biological replicates). Error bars are shown as mean ± SEM.

We also performed EMSA experiments to verify the specificity of the AaERR binding site of AaERR protein in the promoter region of *FAS*. When the biotin-labeled *FAS* probe combined with nucleoprotein overexpressed AaERR-V5 protein, the specific complex band appeared in the gel lane. And this band disappeared after adding the 50 × molar excess competitive unlabeled probe to the labeled probe and nucleoprotein mixture. The AaERR-V5 nucleoprotein was incubated with the anti-V5 antibody on ice for 1–2 h before being mixed with the labeled probe, which resulted in the disappearance of the band shift. Contrary to the addition of the unlabeled probe, the band was notably observed again following the addition of the unlabeled mutated probe. A similar band shift was observed when the anti-IgG antibody was incubated with the nucleoprotein sample, confirming the specificity binding of AaERR-V5/probe and anti-V5 antibody (Fig 7C). Taken together, these experiments conclusively show that AaERR directly interacts with its specific binding site in the *FAS* promoter, affecting expression of this LM gene.

ChIP/qPCR analyses with AaERR-V5 overexpressed Aag2 cells using anti-V5 antibody showed that the promoter fragments of *FAS*, *ACSL* and *ACACA* in the LM pathway were highly enriched (Figs 7D and S3D). Subsequent ChIP with the same antibody, followed by qPCR of their coding regions, showed no enrichment compared to the promoter regions (Figs 7E and S3F). This suggests that AaERR-specific binding motifs are also present in the promoters of *FAS*, *ACSL*, and *ACACA*.

## Discussion

In mosquitoes, 20E plays a major role in regulating PBM reproductive events, and its action is mediated by the cascade of transcription factors, many of which belong to the nuclear receptor family [13,42–44]. In this study, AaERR, a member of NR3B subfamily of NRs, was investigated during reproduction of the mosquito *Ae. aegypti*. We found that *AaERR* gene expression showed a correlation with the titer of 20E. The *AaERR* gene was upregulated in the presence of 20E but downregulated after *EcR* RNAi suggesting the 20E involvement on this gene regulation (Fig 1). Studies in *B. mori* have shown that 20E increases the transcriptional activity of *BmERR*, and the EcRE has been found in the *BmERR* promoter region [45]. Using the cell-transfection assay as well as EMSA, we demonstrate that the effect of 20E on *AaERR* expression was due to the direct binding of EcR to the EcRE in the *AaERR* regulatory region (Fig 2). Thus, these results strongly suggest similar regulation of *ERR* expression in *Ae. aegypti* and *B. mori*, indicating that ERR acts downstream of the 20E/EcR signaling pathway. Moreover, we demonstrate here that EcR mediates the action of 20E directly binding to its specific EcRE binding sites in the *AaERR* gene.

We also found that *AaERR* RNAi resulted in abnormal development of the ovary (Fig 3). ERR-deficiency generated fewer offspring in aphids and influenced the embryonic development of *Drosophila* and silkworm [9,29,31]. Additionally, ERR regulates the reproduction of male insects by affecting sexual behavior and testes development [46,47]. Therefore, it appears that ERR regulates the fecundity in various insects. Taken together, AaERR appear to serve as a key factor during mosquito reproduction under the control of 20E/EcR signaling.

Previous studies have shown that NRs play a central role in controlling metabolic homeostasis in insects [20,48,49]. ERR regulates the expression of the glycolytic genes in *B. mori* [31], *Drosophila* [9], and *Myzus persicae* [29]. Consistent with these observations, we found that AaERR reduced the levels of circulating sugars—glucose, fructose, and trehalose—in the female mosquito FB, which might be caused by directly elevating the expression of glycolytic genes (Fig 4).

Our previous study has confirmed that 20E-EcR-USP signaling induced the catabolism of carbohydrate during mosquito reproduction [6]. However, this study did not explain the

molecular basis of 20E/EcR action on metabolic genes in the mosquito. Here we demonstrate that AaERR acts as a direct regulator of CM, with its expression being controlled by 20E-bound EcR in the mosquito (Fig 6). Likewise, HR38 was reported to mediate the 20E regulation of CM during the reproductive cycle of *Ae. aegypti* [5]. Collectively, these observations in adult mosquito FBs have revealed that 20E/EcR promoted the expression of CM genes by regulating AaERR and HR38. A precise coordination between AaERR and HR38 in regulation of mosquito metabolism remains to be elucidated. Interestingly, recent work in *Drosophila* larvae has shown that EcR works together with ERR in regulating the transcription of CM genes by jointly binding to the same regulatory elements. Expression of these target genes repressed by 20E treatment [32].

In *Drosophila*, ERR has been shown to act as a transcriptional switch regulating lipogenesis and increasing the levels of long-chain fatty acids [33]. Our transcriptomic-base analysis revealed that a substantial number of genes intricately associated with fatty acid synthesis are significantly downregulated in iAaERR mosquitoes at 36 h PBM (Fig 5). Moreover, this depletion of *AaERR* resulted in the decreased size and content of lipid droplets as well as the storage of TAG (Fig 4). Therefore, we speculated that AaERR was mainly involved in the anabolism of lipids in the FB of female mosquitoes at 36 h PBM. Previous studies have demonstrated that the transcription levels of lipid catabolism related enzyme genes in the FB of mosquitoes are significantly upregulated, and the lipid stores are reduced in a 20E/EcR-HNF4 signaling-dependent manner during the PBM stage [7]. Our results suggest that NRs could cooperate with each other to regulate LM homeostasis in the FBs after a blood meal (Fig 7). The details of the interplay between 20E and NRs in regulation of LM require further investigation. We discovered that AaERR exerts contrasting effects on CM and LM, leading us to postulate a regulated equilibrium between these metabolic pathways in mosquito reproductive cycles to maintain homeostasis.

In summary, this study reveals how AaERR coordinates metabolism with the energy requirement during mosquito reproduction in a 20E-dependent manner. As the primary regulator of the reproductive processes during the PBM stage in female mosquitoes, 20E binds to the EcR/USP complex, which in turn targets the *AaERR* promoter EcRE region to upregulate the expression of *AaERR*. Subsequently, AaERR affects the metabolic flux of mosquitoes by directly regulating the expression of genes, such as *GPI* and *PYK* in the CM pathway and *FAS* in the LM pathway. Furthermore, we identified that the regulatory effects of AaERR on metabolism is critical for the fecundity of *Ae. aegypti* females. Taken together, this work provides an important framework for further exploration of the hormonal-metabolic axis in the pathogen-transmitting *Ae. aegypti* mosquito.

## Materials and methods

### Ethics statement

The Institute of Zoology's Animal Care and Use Committee granted approval for all procedures involving vertebrate animals (IOZ20190062).

### Experimental animals and cell culture

*Ae. aegypti* (Liverpool strain) mosquitoes were raised as previously described [38,50]. In brief, mosquitoes were cultured at about 27˚C with a humidity of 80%. Adult mosquitoes were continuously supplied with 10% sucrose solution and water. Subsequently, adult female mosquitoes were provided with a blood meal from specific pathogen-free chickens. *Drosophila* S2 cells were used in the dual luciferase reporter assay, and mosquito Aag2 cells were used in the EMSA and ChIP. S2 and Aag2 cells were kept at 27˚C and cultured in Schneider's insect

culture medium from Sigma, supplemented with 10% and 8% fetal bovine serum (FBS, sourced from Gibco), respectively.

## RNAi experiments, RNA preparation and qRT-PCR analysis

As mentioned previously [51], dsRNAs used in the microinjection experiments were prepared according to the instructions of the T7 RiboMAX Express RNAi System (Promega). The Nanoliter 2000 injector, a device produced by World Precision Instruments, was employed to inject the dsRNAs into the thorax of female mosquitoes. These mosquitoes were anesthetized using cold exposure, and the procedure was carried out within 24 h PE. For *AaERR* RNAi, blood was fed on the third day after injection of dsRNAs, and samples were collected at 36 h PBM.

The FB tissue of ten mosquitoes were homogenized using a homogenizer and lysed with TRIzol reagent (Invitrogen), and total RNAs were extracted as described previously [6]. Fast Quant RT kit (Tiangen) was used to synthesize cDNA. At least four repeated qRT-PCR reactions were performed on the applied biosystems qPCR machine (Thermo Fisher Scientific) using SYBR Green SuperReal PreMix (Tiangen). *Ribosomal protein S7* (*rps7*) was used as the internal control to normalize different samples. The primers used are shown in S1 Table.

## GC-MS, Glycogen and TAG measurements

FBs of six female mosquitoes per sample were ground in the mixed reagent (methanol: ethanol: chloroform = 8:1:1) and incubated at -20˚C for 1 h. Subsequently, samples were centrifuged, and the supernatant was dried to prepare the derivatives using O-methoxyamine hydrochloride (saturated in pyridine) and MSTFA. Finally, following the centrifugation, the supernatant was diluted with n-hexane and transferred to the chromatographic-grade sample bottles for the GC-MS analysis. Temperature steps were as mentioned in prior studies [6]. During the experiment, n-hexane and decanoic acid were used as the solvent and internal reference, respectively.

As reported previously [6,52], glucose reagent (Sigma) was used for glycogen determination. All groups analyzed contained at least six biological repeats, and each sample had six female FBs. After processing and centrifuging, the samples were separated into three parts and transferred to a 96-well plate. Each aliquot was mixed with 1× PBS, glucose, and glucose reagent with amylase separately, and incubated at 37˚C for 30 minutes (min). The reaction was terminated using 12 N sulfuric acid. SpectraMax Plus384 was used to analyze the results. For TAG detection, FBs obtained from six female mosquitoes were processed in a homogenizer using a 100 μl of 1× PBS buffer that contained 0.5% Tween-20. After this, the samples underwent heat treatment at 70˚C for 5 min. In the next steps, the samples were sequentially exposed to triglyceride reagent and free glycerol reagent (both products from Sigma) and analyzed by means of colorimetric determination.

## Glycogen and lipid droplets staining

The histochemical staining and visualization of glycogen and lipid droplets in FBs of mosquitoes were conducted as previously reported [53]. Briefly, the abdomen of female mosquitoes was dissected and fixed with 4% paraformaldehyde at 4˚C for 12 h. Following dehydration with different concentrations of ethanol and embedding in paraffin, samples were cut into 3- to 5-μm sections. Finally, the PAS method (Sigma, 395B) was used to stain the sections, and the glycogen was observed under a Nikon Ni-E microscope. For lipid droplets staining, the FB tissues of the mosquito were dissected and washed two to three times by rinsing in 1× PBS. They were then subjected to a 2 h incubation in Nile Red reagent, then fixed on glass slides

and observed under a Zeiss LSM 710 confocal microscope. Zeiss ZEN software was used to measure the size of lipid droplets.

### *In-vitro* FB culture experiments

To demonstrate the impact of 20E on the expression of the *AaERR* gene in a controlled environment, FBs were extracted from female mosquitoes at 72 h PE. These FBs were then cultured in a medium that contained either 20E or DMSO, the latter serving as a control. First, FBs were incubated in a medium containing $2.5 \times 10^{-1}$ μM 20E for 2 h and then 2 μM 20E for 10 h more. Each group contained four biological repeats.

### Dual luciferase reporter assay and EMSA

Coding regions of *EcR*, *USP*, and *AaERR* were cloned into the pAc5.1b expression vector, and the corresponding promoter regions were cloned into the pGL4.10 vector. Plasmids of interest were introduced into *Drosophila* S2 cells using the Lipofectamine 3000 transfection reagent from Invitrogen. Following transfection by 38 h, 20E was introduced to achieve a terminal concentration of 2 μM, which was sustained for a subsequent 10 h. The resulting luciferase activity was measured using the Dual-Luciferase Reporter Assay System and a GloMax 96 Microplate Luminometer, both from Promega.

In a similar manner, Aag2 cells were transfected with the intended plasmids utilizing the FUGENE 6 transfection reagent. After 48 h, nuclear proteins were isolated using the NE-PER nuclear and cytoplasm extraction kit from Thermo Scientific. EMSA experiments were subsequently carried out in accordance with the procedures outlined in the Pierce Light Shift Chemiluminescent EMSA Kit, also from Thermo Scientific [54]. The extracted nucleoprotein and biotin-labeled probe were incubated in the binding buffer for 30 min at 25˚C, according to the method provided in the kit instructions. $50 \times$ molar excess of the probe without biotin labeling was added to the binding reagent for competition experiments. In the super shift assays, the anti-V5 antibody (Invitrogen) was incubated with the nucleoprotein on ice for 1–2 h prior to mixing with the labeled probes. The reaction mixtures were separated on a 6% native polyacrylamide gel and results were observed on X-ray film. The probes used in this analysis are shown in S1 Table.

### Western blot analysis

To extract total proteins, FBs of ten female mosquitoes were homogenized in a lysis buffer (Beyotime) with protease inhibitors (Pierce) using a grinding rod. Following a minimum of 30 min incubation on ice, the lysate was subjected to separation on an SDS-polyacrylamide gel from Bio-Rad. Following the transfer of proteins onto a PVDF membrane (HYCX), it was incubated with the primary and corresponding secondary antibodies. Results were visualized after the treatment of SuperSignal West Pico Chemiluminescent Substrate (Thermo Fisher Scientific).

### ChIP-qPCR

Aag2 cells were transfected with either pAc5.1b-EcR/USP vectors, pAc5.1b-ERR vector, or the pAc5.1b vector for 48 h. Post transfection, the cells were treated with 1% formaldehyde at room temperature for 10 min. Following this, lysis buffer was used to collect chromatin, as previously described [55,56]. The chromatin fragments obtained after ultrasonication were then immunoprecipitated overnight with 5 μg of antibody (either anti-V5 or anti-IgG) at 4˚C. Both ChIP and input DNA were subsequently purified using the QIAquick Spin PCR Purification Kit (Qiagen, CA) and analyzed via qPCR.

## Bioinformatics analysis

High-quality clean reads were filtered from raw reads obtained through high-throughput sequencing (Illumina NovaSeq sequencing platform) using the software Cutadapter [57] (version 1.11). These clean reads data were then aligned to a reference genome using the hisat2 software [58] (version: 2.0.1-beta). We used feature counts [59] (version: v1.6.0) to calculate the number of reads on genes according to the annotation file of the genome and standardized the number of reads of genes. The standardized method adopted here was FPKM (fragments per kilobase of exon per million reads mapped) [60]. Differences in gene expression levels between iEGFP and iAaERR mosquito samples was assessed using edgeR [61]. Genes with a p value <0.05 and fold change >1.5 or <0.667 were considered significant differential genes. Visualization of these results on a volcano plot was achieved using the online tool at https://www.bioinformatics.com.cn. GO enrichment analysis of upregulated and downregulated genes was performed in an R environment using the David web server (https://david.ncifcrf.gov/tools.jsp). KEGG pathway enrichment analysis was performed using the KOBAS 3.0 web server (http://kobas.cbi.pku.edu.cn/). GO terms and pathways with a p value <0.05 were considered to be significantly enriched. Last, Sankey and dot plots were generated using online tool at https://www.bioinformatics.com.cn.

## Statistical analysis

All statistical evaluations, excluding those related to RNA-Seq data, were conducted using the GraphPad statistical software. The statistical test methods to conduct the significant variation among different treatments are shown in S3 Table. A p value <0.05 was considered indicative of statistical significance.

## Supporting information

**S1 Fig. The RNAi efficiency of *Met*, *EcR*, and *AaERR* genes in *Ae. aegypti*.** The relative mRNA levels of *Met* (A) and *AaERR* (B) after *Met* knockdown (Two-tailed Unpaired t test: ****p < 0.0001; at least three biological replicates). Error bars are shown as mean ± SEM. (C) The relative mRNA level of *EcR* after *EcR* knockdown (Two-tailed Unpaired t test: ****p = 0.0005; at least three biological replicates). Error bars are shown as mean ± SEM. (D) The relative mRNA level of *AaERR* after *AaERR* knockdown (Two-tailed Welch's t test: ****p < 0.0001; at least three biological replicates). Error bars are shown as mean ± SEM. *EGFP* knockdown was used as the control in (A-D).
(TIF)

**S2 Fig. KEGG pathway enrichment analysis of DEGs in iAaERR mosquitoes.** KEGG pathway enrichment analysis was performed on the upregulated DEGs in FBs of iAaERR female mosquitoes.
(TIF)

**S3 Fig. Effect of iAaERR on the regulation of key enzymes in the CM and LM pathways.** (A-B) The relative mRNA levels of CM and LM genes were analyzed using qRT-PCR in iAaERR mosquitoes (Two-tailed Unpaired t test: ****p < 0.0001; Two-tailed Mann-Whitney test: *p = 0.0286; at least three biological replicates). Error bars are shown as mean ± SEM. PFK, phosphofructokinase; PGM, phosphoglucomutase; ACSL, long-chain acyl-CoA synthetase; ACACA, acetyl-CoA carboxylase/biotin carboxylase 1. iEGFP samples were used as control. (C-D) ChIP-qPCR analysis using anti-V5 and anti-IgG antibodies assessed the enrichment of *PFK*, *PGM*, *ACSL*, and *ACACA* promoter fragments in Aag2 cells transfected with pAc5.1b-AaERR-V5. Control cells were transfected with the pAc5.1b vector and

subjected to the same antibody analysis. The graph illustrated the relative fold enrichment for promoter fragments of each gene (Two-tailed Unpaired t test: ****$p < 0.0001$; three biological replicates). Error bars are shown as mean ± SEM. (E-F) ChIP with an anti-V5 antibody followed by qPCR assessed the enrichment of the regulatory regions versus control regions within the coding sequences of *PFK*, *PGM*, *ACSL*, and *ACACA*. (Two-tailed Welch's t test: **$p = 0.0075$ (*PFK*); Two-tailed Unpaired t test: ***$p = 0.0008$ (*PGM*); Two-tailed Welch's t test: **$p = 0.001$ (*ACSL*); Two-tailed Unpaired t test: *$p = 0.0162$ (*ACACA*); three biological replicates). Error bars are shown as mean ± SEM.
(TIF)

**S1 Table. The primers used in this study.**
(DOCX)

**S2 Table. The putative AaERR binding sequence in the promoter region of the CM and LM key enzyme genes in *Ae. aegypti*.**
(DOCX)

**S3 Table. Detailed presentation of two-tailed statistical test results and effect outputs.** KD, knock down; FBs, fat bodies.
(DOCX)

**S4 Table. Transcriptomic changes of differentially expressed CM and LM genes in iAaERR mosquitoes.** ACLY, ATP citrate (pro-S)-lyase; MDH1, malate dehydrogenase; MDH2, malate dehydrogenase; TPI, triosephosphate isomerase; CHK, choline/ethanolamine kinase; IDH, isocitrate dehydrogenase; MINPP1, inositol-polyphosphate phosphatase; INO1, inositol-3-phosphate synthase.
(DOCX)

**S5 Table. Numerical data plotted in graphs.**
(XLSX)

## Author Contributions

**Conceptualization:** Dan-Qian Geng, Xue-Li Wang, Alexander S. Raikhel, Zhen Zou.

**Data curation:** Dan-Qian Geng, Xue-Li Wang, Xiang-Yang Lyu.

**Formal analysis:** Dan-Qian Geng, Xue-Li Wang.

**Funding acquisition:** Xue-Li Wang, Alexander S. Raikhel, Zhen Zou.

**Project administration:** Alexander S. Raikhel, Zhen Zou.

**Resources:** Dan-Qian Geng, Xue-Li Wang.

**Supervision:** Alexander S. Raikhel, Zhen Zou.

**Writing – original draft:** Dan-Qian Geng, Xue-Li Wang.

**Writing – review & editing:** Alexander S. Raikhel, Zhen Zou.

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
