## [Decision Letter · Decision Letter 0]

6 Nov 2023

Dear Dr Zou,

Thank you very much for submitting your Research Article entitled 'Ecdysone-controlled nuclear receptor ERR regulates metabolic homeostasis in the disease vector mosquito Aedes aegypti' to PLOS Genetics.

The manuscript was fully evaluated at the editorial level and by independent peer reviewers. The reviewers appreciated the attention to an important problem, but raised some substantial concerns about the current manuscript. Based on the reviews, we will not be able to accept this version of the manuscript, but we would be willing to review a much-revised version. We cannot, of course, promise publication at that time.

If you decide to revise the manuscript for further consideration at PLOS Genetics, please aim to resubmit within the next 60 days, unless it will take extra time to address the concerns of the reviewers, in which case we would appreciate an expected resubmission date by email to plosgenetics@plos.org.

We are sorry that we cannot be more positive about your manuscript at this stage. Please do not hesitate to contact us if you have any concerns or questions.

Yours sincerely,

Subba Reddy Palli, Ph.D.

Academic Editor

PLOS Genetics

Gregory P. Copenhaver

Editor-in-Chief

PLOS Genetics

Reviewer's Responses to Questions

**Comments to the Authors:**

Reviewer #1: In this study, Gen et al. delved into the functional implications of the estrogen-related receptor (ERR) in the regulation of metabolism in adult female mosquitoes following a blood meal. Their tissue culture and RNAi experiments demonstrated that the upregulation of ERR in blood-fed mosquitoes was triggered by the elevation of 20E after blood feeding. Notably, the knockdown of ERR via RNAi in female mosquitoes elicited significant shifts in various carbohydrate and lipid metabolites, accompanied by significant changes in the expression of key genes involved in these metabolic pathways, resulting in a decrease in mosquito fecundity. This study established ERR as a key player of the 20E regulatory cascade, directly modulating metabolic flux during mosquito reproduction. While the regulatory function of ERR in metabolic homeostasis has been documented in other insect species, this study is particularly significant due to its focus on the metabolic role of ERR in the context of 20E signaling in a vital disease vector. The findings of this study provide significant advancement in the understanding of how metabolism is mobilized to meet the energy demands for mosquito adult reproduction. The conducted experiments were meticulously designed, incorporating appropriate controls and rigorous statistical analyses.

However, an important concern arises regarding the study's emphasis on genes positively regulated by ERR, potentially neglecting those that may be negatively regulated. To provide a more comprehensive understanding, it is imperative to investigate whether any of the 118 upregulated genes following ERR knockdown are associated with metabolism. Additionally, exploring potential ERR binding sites within these upregulated genes would significantly contribute to a more balanced interpretation of the underlying molecular mechanisms.

Other points:

1. Line 85. Rephrase this sentence to enhance clarity.

2. Fig. 1D. Specify the time point for the measurement of the ERR transcript after the knockdown of EcR.

3. Fig. 2. Rearrange the legend to place the description of error bars after the statistical data for improved comprehension.

4. EMSA results (Figs. 2, 6, and 7). Rectify the description of V5 antibody and the labeled AaERR (cold) probe as negative controls while the mutant probe and IgG as positive controls. They were introduced to validate the specific binding of AaERR to its binding site, not as experimental controls.

5. Line 180. Correct the terminology to reflect that the reporter assay measured luminescence, not fluorescence.

6. Line 251. Clarify that the qPCR measured only 4 genes, not all genes.

7. Lines 279-280. Provide a precise description that the addition of anti-IgG did not affect the probe-protein interaction or the appearance of the shift band.

8. Line 300. Specify that nuclear proteins were preincubated with the anti-V5 antibody for clarity.

9. Lines 301-304. Elaborate on the opposite effect observed, providing explicit details to avoid ambiguity.

10. Line 309. The statement is inaccurate. Not all members of the 20E regulatory cascades are nuclear receptors.

11. Fig 5. Explain the concept of Gene Ratio to facilitate better comprehension.

12. Consider adding a dataset to report the transcriptomic changes of annotated CM and LM genes in iAaERR mosquitoes to provide additional context and support.

13. Provide an explanation of how the downregulation of very long-chain acyl-CoA dehydrogenase (ACADVL), an enzyme required for fatty acid oxidation, fits into the observed decrease in the levels of TAG in iAaERR mosquitoes.

Reviewer #2: Working on the mosquito Aedes aegypti, Geng and colleagues found that ERR knockdown resulted in deficient phenotypes of ovary as well as the metabolism of carbohydrates and lipids during the gonadotropic cycle. ERR transcription was directly activated by 20E via EcR binding to the EcRE in ERR promoter. They also found that ERR directly activated the expression of metabolic genes GPI, PYK and FAS by binding to the corresponding ERR-responsive motif in the promoters of these genes. Overall, the study represents a new and interesting finding in hormonal regulation of female reproduction. The data are comprehensive and presented in a logical sequence. The manuscript is well written. However, there are some concerns that need to be addressed.

Specific comments:

1. The developmental profile of 20E titer could be added in Fig 1A. ERR distributes in both cytoplasm and nuclei from PBM 24h to 48h, but mainly in nuclei at 72h (Fig 1B). Authors should discuss the dynamic of subcellular localization of ERR, as well as its relation with ERR regulation of carbohydrate and lipid metabolism during the mosquito reproduction.

2. Please indicate pAc5.1b-USP-V5 in Fig 2 and its legend.

3. The tissues (fat body or hemolymph) should be addressed for glucose, fructose, and trehalose measurement in Fig 4A and4 C as well as the Results and figure legends.

4. Among ME genes, ERR directly bound to the promoters of GPI and PYK and activated their expression. How about other CM pathway genes like PFK and PGM? Authors should discuss the potential regulation of GPI and PYK by ERR.

5. Similarly, ERR directly bound to the promoter of FAS and activated its expression, whereas ACSL and ACACA were not directly regulated by ERR. This should be discussed too.

6. Additional bands were seen when IgG was present in EMSA of Fig 6D but not Fig 7C. In addition to luciferases assay and EMSA, ChIP could be a good plus to investigate the binding of EcR/USP and ERR to the promoters.

**Have all data underlying the figures and results presented in the manuscript been provided?**

Reviewer #1: **No: **The dataset of transcriptomic analysis is missing.

Reviewer #2: None

PLOS authors have the option to publish the peer review history of their article (what does this mean?). If published, this will include your full peer review and any attached files.

Reviewer #1: No

Reviewer #2: No

---

## [Decision Letter · Decision Letter 1]

20 Feb 2024

Dear Dr Zou,

We are pleased to inform you that your manuscript entitled "Ecdysone-controlled nuclear receptor ERR regulates metabolic homeostasis in the disease vector mosquito Aedes aegypti" has been editorially accepted for publication in PLOS Genetics. Congratulations!

Yours sincerely,

Subba Reddy Palli, Ph.D.

Academic Editor

PLOS Genetics

Gregory P. Copenhaver

Editor-in-Chief

PLOS Genetics

Comments from the reviewers (if applicable):

Reviewer's Responses to Questions

**Comments to the Authors:**

Reviewer #2: The authors have satisfactorily addressed my concerns raised in initial review. Additional experiments have been done and included in Fig. 1A, Fig. 2C-D, Fig. 6E-F, Fig. 7D-E, and Fig. S3C-F. The manuscript is considerably improved. I recommend acceptance for publication in the current form.

**Have all data underlying the figures and results presented in the manuscript been provided?**

Reviewer #2: None

PLOS authors have the option to publish the peer review history of their article (what does this mean?). If published, this will include your full peer review and any attached files.

Reviewer #2: No

**Data Deposition**

http://datadryad.org/submit?journalID=pgenetics&manu=PGENETICS-D-23-01119R1

**Press Queries**

---

## [Editor Report · Acceptance letter]

6 Mar 2024

PGENETICS-D-23-01119R1 

Ecdysone-controlled nuclear receptor ERR regulates metabolic homeostasis in the disease vector mosquito Aedes aegypti 

Dear Dr Zou, 

We are pleased to inform you that your manuscript entitled "Ecdysone-controlled nuclear receptor ERR regulates metabolic homeostasis in the disease vector mosquito Aedes aegypti" has been formally accepted for publication in PLOS Genetics! Your manuscript is now with our production department and you will be notified of the publication date in due course.

With kind regards,

Anita Estes

PLOS Genetics

On behalf of:
